# Gaussian Processes with Bayesian Inference of Covariate Couplings

## Abstract

Gaussian processes are powerful probabilistic models that are often coupled with Automatic Relevance Determination (ARD) capable of uncovering the importance of individual covariates. We develop covariances characterized by affine transformations of the inputs, formalized via a precision matrix between covariates, which can uncover covariate couplings for enhanced interpretability. We study a range of couplings priors from Wishart to Horseshoe and present fully Bayesian inference of such precision matrices within sparse Gaussian process. We demonstrate empirically the efficacy and interpretability of this approach.

## 1 Introduction

Statistical models based on Gaussian Processes (GPs) offer attractive modeling choices for various quantitative sciences due to their ability to impose functional priors with certain desired characteristics and to carry out principled uncertainty quantification (Rasmussen & Williams, 2006). Modeling and inference of GPs has evolved significantly in the directions of scalability for large data (Cutajar et al., 2017; Hensman et al., 2013; Wilson & Nickisch, 2015), deep learning (Damianou & Lawrence, 2013; Wilson et al., 2016; Salimbeni & Deisenroth, 2017), and generality with autodiff frameworks (Krauth et al., 2017; Matthews et al., 2017).

The choice of the covariance (kernel) function plays a crucial role in specifying the function space induced by GPs. This choice is often overlooked by opting for the reputable "default" exponential ARD covariances (Neal, 1996), which capture the importance of each covariate, but also assumes an *axis-aligned* anisotropic data structure, blind to covariate couplings (Matérn, 1960).

In contrast, *affine* anisotropic covariances are able to explicitly consider the linear dependencies between covariates (Matérn, 1960; Poggio & Girosi, 1990), which is a common feature of real-world data, via the *precision* matrix $\mathbf{\Lambda}$ of the Mahalanobis distance $(\mathbf{x} - \mathbf{x}')^\top \mathbf{\Lambda} (\mathbf{x} - \mathbf{x}')$. A seminal work of Vivarelli & Williams (1998) proposes a parameterization of the precision based on Principal Component Analysis (PCA), while Titsias & Lazaro-Gredilla (2013) apply mean-field variational inference over factors of such a precision matrix $\mathbf{\Lambda}$. Relevant works on affine-covariances GPs include non-stationary extensions (Paciorek & Schervish, 2003), and applications to imaging (Kalaitzis, 2009) and material sciences (Noack et al., 2020).

In this paper, our goal is to revitalise Mahalanobis distance-based covariances as a more interpretable and general alternative to "diagonal" ARD covariances, whereby we are able to uncover covariate couplings. This is illustrated in Fig. 1, where we refer to these more general types of covariance functions as Automatic Coupling Determination (ACD) covariances. We study a fully Bayesian scalable formulation of GPs, where we carry out inference over the matrix $\mathbf{\Lambda}$, thus obtaining posterior distributions over covariate couplings.

Our contributions are as follows: (i) a GP model that determines covariate couplings through the analysis of the matrix $\mathbf{\Lambda}$; (ii) an analysis of sparsity-inducing priors for the matrix $\mathbf{\Lambda}$ from Wishart, Laplace and Horseshoe families; (iii) a demonstration of the enhanced explainability of ACD covariances compared to ARD covariances; (iv) the development of a fully Bayesian Markov chain Monte Carlo (MCMC) inference scheme of the couplings; and (v) an empirical demonstration of the usefulness of ACD covariances.

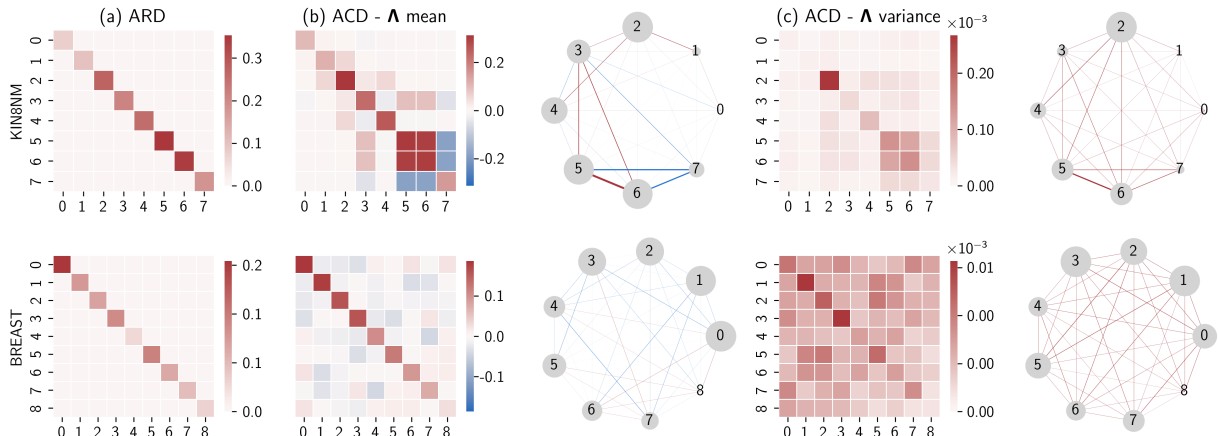

**Figure 1: The Automatic Coupling Determination (ACD) covariance reveals rich predictive covariate couplings.** Comparison between ARD diagonal precisions $\mathbf{\Sigma}^{-1} = \mathrm{diag}(\boldsymbol{\ell}^{-2})$ (*a*) and ACD precision matrix $\mathbf{\Lambda}$ mean (*b*) and variance (*c*) with graph illustrations. We assume an element-wise Normal prior on $\mathbf{\Lambda}$. The ACD covariance detects that the covariates (5,6,7) are close to redundant on the `kin8nm` dataset.

## 2 Background

We consider supervised learning problems with $N$ input-label pairs $\{\mathbf{X}, \mathbf{y}\} = \{(\mathbf{x}_n, y_n)\}_{n=1}^{N}$, with $\mathbf{x}_n \in \mathbb{R}^D$ and $y_n \in \mathbb{R}$ using Gaussian processes (Rasmussen & Williams, 2006). Let $\mathbf{f} = (f(\mathbf{x}_1), ..., f(\mathbf{x}_N)) \in \mathbb{R}^N$ be unknown latent variables of inputs $\mathbf{X} = (\mathbf{x}_1, ..., \mathbf{x}_N)$, and $\prod_{i=1}^{N} p(y_n|f_n)$ be an i.i.d likelihood.

### 2.1 Gaussian process priors

By imposing a GP prior

$$f(\mathbf{x}) \sim \mathcal{GP}(0, k) \tag{1}$$

on the latent variables $\mathbf{f}$, we are assuming that they are jointly Gaussian (Rasmussen & Williams, 2006). The covariance of the multivariate Gaussian over $\mathbf{f}$, which determines the properties of the functions that can be drawn from the prior, is the kernel function $k(\mathbf{x}, \mathbf{x}'; \boldsymbol{\theta})$, where $\boldsymbol{\theta}$ are hyper-parameters. The prior over $\mathbf{f}$ is then $p(\mathbf{f}|\boldsymbol{\theta}) = \mathcal{N}(\mathbf{0}, \mathbf{K}_{\mathrm{xx}|\boldsymbol{\theta}})$, where $\mathbf{K}_{\mathrm{xx}|\boldsymbol{\theta}}$ is the $N \times N$ covariance matrix obtained by evaluating $k(\mathbf{x}, \mathbf{x}'; \boldsymbol{\theta})$ at all input pairs $\{\mathbf{x}, \mathbf{x}'\}$. For simplicity, we assume zero-mean GPs and omit the conditioning on $\mathbf{X}$.

The posteriors over $\mathbf{f}$ at inputs $\mathbf{x}_*$, and inference or optimization over $\boldsymbol{\theta}$ is based on the analysis of the joint

$$p(\mathbf{y}, \mathbf{f}, \boldsymbol{\theta}) = p(\mathbf{y}|\mathbf{f})p(\mathbf{f}|\boldsymbol{\theta})p(\boldsymbol{\theta}). \tag{2}$$

With Gaussian likelihoods it is possible to marginalize out $\mathbf{f}$ leading to a Gaussian $p(\mathbf{y}, \boldsymbol{\theta}) = p(\mathbf{y}|\boldsymbol{\theta})p(\boldsymbol{\theta})$. With non-Gaussian likelihoods further complications arise due to the lack of conjugacy (Williams & Barber, 1998; Opper & Winther, 2000).

An overarching issue with GP models is scalability, as these models generally require costly $\mathcal{O}(N^3)$ operations involving $\mathbf{K}_{\mathrm{xx}|\boldsymbol{\theta}}$ inverses. Linearization techniques based on random features (Rahimi & Recht, 2008) were proposed in Lázaro-Gredilla et al. (2010), and they were later developed to operate with mini-batches within stochastic gradient optimization and to deep GPs (Cutajar et al., 2017). Sparsification techniques based on inducing points (Williams & Seeger, 2000; Snelson & Ghahramani, 2005) were later embedded within a variational formulation (Titsias, 2009), and they were extended to mini-batching (Hensman et al., 2013; Krauth et al., 2017). In this paper we consider sparse GPs, and in particular their fully Bayesian version presented in Rossi et al. (2021), where all variables are treated in a Bayesian way and inference is carried out using stochastic gradient MCMC (Chen & Zhang, 2004).

## 2.2 Fully Bayesian sparse GPs

We focus on the Bayesian sparse Gaussian process (BSGP) framework (Rossi et al., 2021), but the ACD covariance specifications apply in general to any GP implementations. In sparse GPs, we introduce a set of $M$ inducing variables $\mathbf{u} = (u_1, ..., u_M)$ at inducing inputs $\mathbf{Z} = \{\mathbf{z}_1, ..., \mathbf{z}_M\}$, such that $u_m = f(\mathbf{z}_m)$ (Candela & Rasmussen, 2005). The inducing variables are assumed to follow the original GP, yielding a joint GP prior

$$p(\mathbf{f}, \mathbf{u}|\mathbf{X}, \mathbf{Z}, \boldsymbol{\theta}) = p(\mathbf{f}|\mathbf{u}, \mathbf{X}, \mathbf{Z}, \boldsymbol{\theta})p(\mathbf{u}|\mathbf{Z}, \boldsymbol{\theta}) \tag{3}$$

$$p(\mathbf{u} \,|\, \mathbf{Z}, \boldsymbol{\theta}) \sim \mathcal{N}(\mathbf{0}, \mathbf{K}_{\mathrm{zz}|\boldsymbol{\theta}}) \tag{4}$$

$$p(\mathbf{f} \,|\, \mathbf{u}, \mathbf{X}, \mathbf{Z}, \boldsymbol{\theta}) \sim \mathcal{N}(\mathbf{A}\mathbf{u}, \mathbf{K}_{\mathrm{xx}|\boldsymbol{\theta}} - \mathbf{A}\mathbf{K}_{\mathrm{xz}|\boldsymbol{\theta}}^{\top}), \tag{5}$$

where $\mathbf{A} = \mathbf{K}_{\mathrm{xz}|\boldsymbol{\theta}}\mathbf{K}_{\mathrm{zz}|\boldsymbol{\theta}}^{-1}$. This augmented model can be used for modeling tasks by introducing a likelihood $p(\mathbf{y}|\mathbf{f})$. We assign priors over all remaining variables $p_{\boldsymbol{\psi}}(\boldsymbol{\theta})$ and $p_{\boldsymbol{\xi}}(\mathbf{Z})$, notably including inducing locations $\mathbf{Z}$ and kernel hyperparameters $\boldsymbol{\theta}$ (Rossi et al., 2021). The joint becomes

$$p(\boldsymbol{\theta}, \mathbf{Z}, \mathbf{u}, \mathbf{f}, \mathbf{y}|\mathbf{X}) = p_{\boldsymbol{\psi}}(\boldsymbol{\theta})p_{\boldsymbol{\xi}}(\mathbf{Z})p(\mathbf{f}, \mathbf{u}|\mathbf{X}, \mathbf{Z}, \boldsymbol{\theta})p(\mathbf{y}|\mathbf{f}). \tag{6}$$

We can use variational formulations to integrate out $\mathbf{f}$ to obtain an objective that factorizes across data. This allows parameter inference over $\boldsymbol{\Psi} \stackrel{\text{def}}{=} \{\mathbf{u}, \mathbf{Z}, \boldsymbol{\theta}\}$ with scalable MCMC based on stochastic gradients (Chen & Zhang, 2004).

## 3 Bayesian inference of covariate couplings

In this section, after briefly discussing covariances with Automatic Relevance Determination (ARD) (MacKay, 1995; Neal, 1996), which induce some scaling of individual covariates, we present an extension involving an affine transformation of the covariates revealing couplings among these. We discuss how this is achieved by introducing a Mahalanobis distance among inputs with a precision matrix $\boldsymbol{\Lambda}$, and we show how to treat this in a Bayesian way by imposing matrix-variate and sparsity-inducing element-wise priors. We term this type of covariance Automatic Coupling Determination (ACD).

### 3.1 Automatic relevance determination

The design of covariance functions for GP models is an important part of the modeling process. Considering the space of functions $f : \mathbb{R}^D \mapsto \mathbb{R}$, the choice of a covariance $\mathrm{cov}[f(\mathbf{x}), f(\mathbf{x}')] = k(\mathbf{x}, \mathbf{x}'; \boldsymbol{\theta})$ determines the prior distribution over $f$ before observing data. A common choice is the Gaussian covariance function (Radial Basis Function (RBF)):

$$k_{\mathrm{RBF}}(\mathbf{x}, \mathbf{x}'; \boldsymbol{\theta}) \propto \exp\left(-\frac{1}{2}d^2(\mathbf{x}, \mathbf{x}'; \boldsymbol{\theta})\right), \tag{7}$$

where $d(\mathbf{x}, \mathbf{x}'; \boldsymbol{\theta})$ is a parametric distance function between inputs $\mathbf{x}$ and $\mathbf{x}'$. This covariance imposes a prior over infinitely differentiable (smooth) functions. Other common covariance functions based on the distance $d(\mathbf{x}, \mathbf{x}'; \boldsymbol{\theta})$ include the Matérn covariance, exponential, arc-cosine; see, e.g., Shawe-Taylor & Cristianini (2004) for an in-depth treatment.

The simplest distance form

$$d^2_{\mathrm{ISOTROPIC}}(\mathbf{x}, \mathbf{x}'; \boldsymbol{\theta}) = \frac{1}{\ell^2}(\mathbf{x} - \mathbf{x}')^{\top}(\mathbf{x} - \mathbf{x}') \tag{8}$$

induces an isotropic covariance, as all input features are scaled by the same length-scale parameter $\ell$ and contribute equally to the distance, which assumes *spherical* data.

Another choice increasing model flexibility introduces covariate-specific length-scales parameters,

$$d^2_{\mathrm{ARD}}(\mathbf{x}, \mathbf{x}'; \boldsymbol{\theta}) = (\mathbf{x} - \mathbf{x}')^{\top}\boldsymbol{\Sigma}^{-1}(\mathbf{x} - \mathbf{x}') \tag{9}$$

with $\boldsymbol{\Sigma} = \mathrm{diag}(\ell_1^2, \ldots, \ell_D^2)$. This choice gives rise to covariances suitable for ARD (MacKay, 1995; Neal, 1996).

Intuitively, this definition is built on the assumption that if a dimension $d$ has a small value of the associated length-scale $\ell_d$ small changes in the covariate would lead to large responses in the target. The covariance induced by this choice is anisotropic with an axis-aligned metric acting as a scaling of individual covariates.

## 3.2 Automatic coupling determination

The family of ARD covariances allows GP models to yield non-parametric and probabilistic mappings from inputs to labels, while simultaneously determining the importance of each covariate if $\ell_d$'s are optimized or inferred. In this paper, we do not limit ourselves to assessing the relevance of each input covariate, but to automatically discover couplings among these in a general way which can be readily applied to any distance-based covariance function.

We replace the diagonal matrix $\boldsymbol{\Sigma}^{-1}$ containing the inverse length-scales with a full Positive Semi-Definite (PSD) precision matrix $\boldsymbol{\Lambda} = \boldsymbol{\Sigma}^{-1}$ in the calculation of distances,

$$d_{\mathrm{ACD}}^2(\mathbf{x}, \mathbf{x}'; \boldsymbol{\theta}) = (\mathbf{x} - \mathbf{x}')^\top \boldsymbol{\Lambda}(\mathbf{x} - \mathbf{x}') \tag{10}$$

$$= \sum_{i,j}^D \Lambda_{ij}(x_i - x_j')^2, \tag{11}$$

yielding the so-called Mahalanobis distance (Titsias & Lazaro-Gredilla, 2013), which can be interpreted as a distance obtained after an affine transformation (rotation and stretching) of the inputs by the identity (Matérn, 1960; Vivarelli & Williams, 1998; Kalaitzis, 2009)

$$d(\boldsymbol{\Lambda}^{\frac{1}{2}}\mathbf{x}, \boldsymbol{\Lambda}^{\frac{1}{2}}\mathbf{x}'; \mathbf{I}) = d(\tilde{\mathbf{x}}, \tilde{\mathbf{x}}'; \boldsymbol{\Lambda}). \tag{12}$$

If the underlying distribution of the inputs $\mathbf{x}$ is Gaussian, this operation produces an implicit *whitening* of the input data yielding $\tilde{\mathbf{x}}$. While the quadratic form in Eq. 10 has an additive form (Eq. 11), the induced functions do not lend themselves to an additive function interpretation (Vivarelli & Williams, 1998; Duvenaud et al., 2011).

We notice that if the precision matrix has zero elements $\Lambda_{ij} = 0$, the distance function ignores the coupling between covariates $i$ and $j$ in the calculation of pairwise distances among inputs.

**Discriminative vs Generative modeling** The parameterization of ACD covariances has apparent connections with Markov Random Fields (MRFs) (cf. Murphy (2023)), whereby the matrix $\boldsymbol{\Lambda}$ is used to specify an adjacency structure for a set of $D$ random variables $\{X_1, \ldots, X_D\}$. MRFs offer the possibility to verify conditional independence properties of groups of random variables based on the analysis of $\boldsymbol{\Lambda}$, while placing no other assumptions on their underlying distribution. While it is tempting to think of the ACD parameterization of the covariance function as something to be used to draw conclusions on conditional independence among covariates, we are effectively not modeling the distribution of these. Instead, we are pushing $\boldsymbol{\Lambda}$ directly in the definition of the GP prior $p(\mathbf{f}|\boldsymbol{\Lambda})$. Therefore $\boldsymbol{\Lambda}$ assumes the interpretation of a precision matrix inducing an affine transformation of the input, which is optimized or inferred based on the marginal likelihood (or a lower bound thereof). Thus the focus is on performing optimization or inference of $\boldsymbol{\Lambda}$ to accurately modeling the labels, with the intention of obtaining some indication of the predictive power of couplings of covariates. We leave the modeling of the input through MRFs as an interesting avenue for future work.

## 3.3 Precision parameterizations

The precision matrix $\boldsymbol{\Lambda}$ in the ACD covariance needs to be symmetric and PSD. The PSD constraint in the ARD covariance is easy to satisfy, since working with a diagonal version of $\boldsymbol{\Lambda}$ only requires to have non-negative elements on its diagonal and consequently, a log-transformation of the length-scales is sufficient.

**Table 1:** Summary of precision priors and the range of hyperparameters studied.

| Prior | Definition | Parameters | Log pdf |
|---|---|---|---|
| Wishart | $p(\mathbf{\Lambda}) = \mathcal{W}(\mathbf{V}, K)$ | $K = D, \mathbf{V} = K^{-1}\mathbf{I}_D$ | $\log C - \sum_d \log \|\mathbf{L}_{dd}\| - \frac{1}{2}\mathrm{Tr}[K\mathbf{\Lambda}]$ |
| Inverse Wishart | $p(\mathbf{\Lambda}) = \mathcal{IW}(\mathbf{V}, K)$ | $K = D, \mathbf{V} = \mathbf{I}_D$ | $\log C - (2K + 1)\sum_d \log |\mathbf{L}_{dd}| - \frac{1}{2}\mathrm{Tr}[\mathbf{V}\mathbf{\Lambda}^{-1}]$ |
| Laplace | $p(\mathbf{\Lambda}_{ij}) = \mathcal{L}(m, b)$ | $m = 0, b \in \{0.01, 0.1, 1\}$ | $\log C - \frac{1}{b}\|\mathbf{\Lambda}_{ij} - m\|_1$ |
| Horseshoe | $p(\mathbf{\Lambda}_{ij}) = \mathcal{HS}(\tau)$ | $\tau \in \{0.01, 0.1, 1\}$ | $\log C + \frac{1}{2\tau^2}\mathbf{\Lambda}_{ij}^2 + \log E_1\left(\frac{1}{2\tau^2}\mathbf{\Lambda}_{ij}^2\right)$ |

### 3.3.1 Lower triangular factorization

In the case of the ACD covariance, optimization or inference of $\mathbf{\Lambda}$ needs to be performed while preserving the PSD constraint, so that it is straightforward to operate with unconstrained optimization/MCMC sampling. Among all factorizations that can be used to express $\mathbf{\Lambda}$, following (Kalaitzis, 2009), a natural parameterization is via the lower-trangular matrix $\mathbf{L}$,

$$\mathbf{\Lambda} = \mathbf{L}\mathbf{L}^\top. \tag{13}$$

This allows us to directly optimize or sample $\mathbf{L}$ element-wise and recover $\mathbf{\Lambda}$. In addition, this parameterization has some computational advantages in calculating Jacobians which are useful within MCMC, as discussed shortly.

### 3.3.2 Low-rank factorizations

The increased flexibility offered by the ACD formulation comes at a computational cost, which we need to deal with: going from learning $\boldsymbol{\ell} \in \mathbb{R}^D$ length-scales in the ARD covariance to learning a full $\mathbf{\Lambda} \in \mathbb{R}^{D \times D}$ matrix. This is why, for problems where the dimensionality $D$ is high but we are at the same time interested in obtaining an informative precision matrix recovering the underlying structure among the $D$ features, we tackle this problem with PCA, similarly to Vivarelli & Williams (1998) and Paciorek & Schervish (2003). Focusing on the ACD distance, by applying a projection to the difference between data samples, we obtain:

$$(\mathbf{x} - \mathbf{x}')^\top \mathbf{P}_d \mathbf{\Lambda}_d \mathbf{P}_d^\top (\mathbf{x} - \mathbf{x}') \tag{14}$$

where $\mathbf{P}_d$ is the $\mathbb{R}^{D \times d}$ matrix obtained from the eigendecomposition of the empirical covariance matrix

$$\mathbf{\Sigma} = \frac{1}{N}\mathbf{X}_c^\top \mathbf{X}_c = \mathbf{P}\mathbf{S}\mathbf{P}^\top, \tag{15}$$

and $\mathbf{X}_c$ is the centered $\mathbb{R}^{N \times D}$ input matrix. To obtain $\mathbf{P}_d$ we select the $d < D$ columns of $\mathbf{P}$ corresponding to the $d$ highest eigenvalues from $\mathbf{S}$. A sample $\mathbf{x} \in \mathbb{R}^D$ can be projected down to $\mathbb{R}^d$ through $\mathbf{P}_d^\top \mathbf{x}$. As a result, we learn a projected version $\mathbf{\Lambda}_d$ in this latent representation of the full precision matrix. By applying the projection in Eq. 14 we recover the precision matrix in the original space.

## 4 Priors over $\mathbf{\Lambda}$

As a consequence of adopting the BSGP framework we need to specify a prior $p_\psi(\boldsymbol{\theta})$ over covariance hyperparameters $\boldsymbol{\theta}$. Dealing with the ACD covariance, the prior is separately placed over both the marginal variance parameter $\sigma_f^2$ and on the precision matrix $\mathbf{\Lambda}$. While the first is simply a LogNormal distribution with a fixed mean and variance, the prior distribution over the precision matrix $\mathbf{\Lambda}$ requires a deeper understanding. First of all, the Cholesky parameterization $\mathbf{\Lambda} = \mathbf{L}\mathbf{L}^T$ in the context of MCMC sampling introduces a change of variable. We impose a prior probability over a non-linear transformation of $\mathbf{L}$, while this is the variable that is actually sampled together with $\mathbf{U}$, $\mathbf{Z}$ and $\sigma_f^2$.

The change of measure induced by the change of variables, requires the determinant of the Jacobian $\mathcal{J}$:

$$p(\mathrm{vec}\,\mathbf{L}) = p(\mathrm{vec}\,\mathbf{\Lambda})\big|\mathcal{J}(\mathrm{vec}\,\mathbf{\Lambda}, \mathrm{vec}\,\mathbf{L})\big| \tag{16}$$

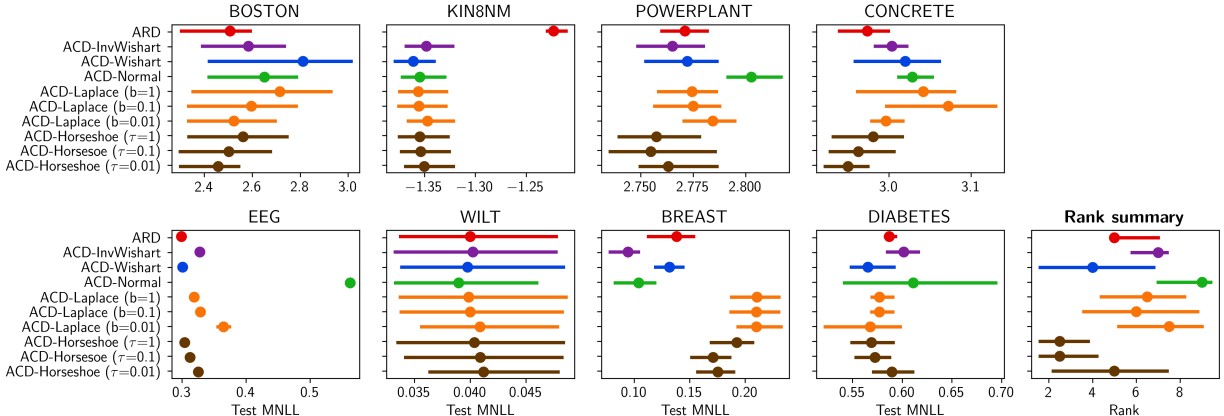

**Figure 2: The** ACD **covariances significantly outperform** ARD **ones on select datasets, while being competetive throughout.** Test mean negative loglikelihood (MNLL) on both UCI regression benchmarks (*top*) and classification (*bottom*) benchmarks with $20\% - 80\%$ error quantiles (lower is better), and rank summaries (*bottom right*).

For the lower-triangular parameterization, the determinant of the Jacobian takes a particularly convenient form, which can be computed linearly in $D$ (Magnus & Neudecker, 1980):

$$\log\left|\mathcal{J}(\text{vec }\mathbf{\Lambda}, \text{vec }\mathbf{L})\right| = \log 2^D \prod_d (\mathbf{L}_{dd})^{D-d+1}. \tag{17}$$

We have identified two different families of priors $p(\mathbf{\Lambda})$: (1) matrix-variate distributions over $\mathbf{\Lambda}$ and (2) factorized scalar distributions over the single entries of the precision $\mathbf{\Lambda}$ defined as $p(\mathbf{\Lambda}) = \prod_{ij} p(\mathbf{\Lambda}_{ij})$.

## 4.1 Matrix-variate priors

**Wishart prior** When dealing with matrix-valued distributions over PSD matrices, a natural probability distribution to consider is the Wishart distribution. Beside being defined over symmetric PSD matrices, the Wishart prior is a conjugate distribution of precision matrices. Considering $\mathbf{\Lambda} \in \mathbb{R}^{D \times D}$ the probability density function can be expressed as:

$$p(\mathbf{\Lambda}) = \mathcal{W}(\mathbf{\Lambda}|\mathbf{V}, K)$$
$$= C|\mathbf{\Lambda}|^{-\frac{1}{2}(K-D-1)} \exp\left(-\frac{1}{2}\text{Tr}(\mathbf{V}^{-1}\mathbf{\Lambda})\right), \tag{18}$$

where $C = (2^{KD}|\mathbf{V}|^{K/2}\Gamma_D(K/2))^{-1}$ is a constant term, $\mathbf{V}$ is the scale matrix and $K \geq D$ is the degrees of freedom parameter. The Bartlett decomposition proves that imposing independent Gaussian priors on the columns of the lower-triangular matrix $\mathbf{L} = (\mathbf{l}_1, ..., \mathbf{l}_D)$ as $p(\mathbf{l}_i) = \mathcal{N}(\mathbf{0}, \mathbf{V})$ is equivalent to a Wishart distribution over $\mathbf{L}\mathbf{L}^T$ as $\mathcal{W}(\lambda\mathbf{I}_D, K)$. We choose $K = D$ degrees of freedom and $\mathbf{V} = D^{-1}\mathbf{I}_D$, such that the expected precision $\mathbb{E}[\mathbf{\Lambda}] = \mathbf{I}_D$ is identity.

**Inverse Wishart** Another prior over PSD matrices related to the Wishart distribution is the inverse Wishart. An interesting interpretation stems from the interpretation of $\mathbf{\Lambda}$ as a covariance matrix in the Fourier domain when Bochner's theorem is applied:

$$k_{\text{RBF-ACD}}(\mathbf{x}_i, \mathbf{x}_j; \sigma_f^2, \mathbf{\Lambda}) \tag{19}$$
$$= \mathbb{E}_{\boldsymbol{\mu}, b} \sqrt{2}\sigma_f \cos(\boldsymbol{\mu}^T\mathbf{x}_i + b) \cdot \sqrt{2}\sigma_f \cos(\boldsymbol{\mu}^T\mathbf{x}_j + b),$$
$$\boldsymbol{\mu} \sim \mathcal{N}(\mathbf{0}, \mathbf{\Lambda}), b \sim \text{Unif}[0, 2\pi]. \tag{20}$$

Therefore, apart from viewing $\boldsymbol{\Lambda}$ as the precision matrix of the kernel, it can also be seen as a covariance matrix in the frequency domain, which offers a motivation for using such a prior

$$
\begin{aligned}
p(\boldsymbol{\Lambda}) &= \mathcal{IW}(\boldsymbol{\Lambda}|\mathbf{V}, K) \\
&= C|\boldsymbol{\Lambda}|^{-\frac{1}{2}(K+D+1)} \exp\left(-\frac{1}{2}\mathrm{Tr}(\mathbf{V}\boldsymbol{\Lambda}^{-1})\right),
\end{aligned}
\tag{21}
$$

where $C = (2^{KD/2}|\mathbf{V}|^{-(K/2)}\Gamma_D(K/2))^{-1}$. We set $K = D$ and $\mathbf{V} = \mathbf{I}_D$. The inverse Wishart view can translate into more efficient random Fourier approximations.

## 4.2  Sparsity-inducing priors

Moving away from matrix-variate distributions, it is possible to encourage sparsity in $\boldsymbol{\Lambda}$ with an element-wise prior. Since we might be interested in promoting sparsity in recovering covariance couplings to a different degree than in the contribution of individual covariates, we separate the prior over the elements of $\boldsymbol{\Lambda}$ as follows:

$$
\begin{aligned}
p(\boldsymbol{\Lambda}) &= p(\boldsymbol{\Lambda}^{\llcorner}) \cdot p(\mathrm{diag}\,\boldsymbol{\Lambda}) \\
&= \prod_{i,j|i\neq j} p(\boldsymbol{\Lambda}_{ij}) \prod_i p(\boldsymbol{\Lambda}_{ii}),
\end{aligned}
\tag{22}
$$

where $\boldsymbol{\Lambda}^{\llcorner}$ and $\mathrm{diag}\,\boldsymbol{\Lambda}$ are the off-diagonal elements and the $\mathbb{R}^D$ array of the diagonal elements of $\boldsymbol{\Lambda}$, respectively. In this work, we assume a weakly informative Gaussian prior on the diagonal of $\boldsymbol{\Lambda}$, while we study different sparsity-promoting prior distributions for $\boldsymbol{\Lambda}^{\llcorner}$, as discussed next.

**Laplace.** A natural way to promote sparse solutions is L1-regularization (cf. graphical lasso in Friedman et al. (2008)), which is equivalent to a Laplace prior. The expression in (22) becomes:

$$
p(\boldsymbol{\Lambda}) = \prod_{i,j|i\neq j} \mathcal{L}(\boldsymbol{\Lambda}_{ij}|m, b) \prod_i \mathcal{N}(\boldsymbol{\Lambda}_{ii}|\mu, \sigma^2),
\tag{23}
$$

where

$$
\mathcal{L}(\boldsymbol{\Lambda}_{ij}|m, b) = C_1 \exp\left(-\frac{1}{b}||\boldsymbol{\Lambda}_{ij} - m||_1\right)
\tag{24}
$$

$$
\mathcal{N}(\boldsymbol{\Lambda}_{ii}|\mu, \sigma^2) = C_2 \exp\left(-\frac{1}{2\sigma^2}\left(\boldsymbol{\Lambda}_{ii} - \mu\right)^2\right),
\tag{25}
$$

where $C_1$ and $C_2$ are the normalizing constants. We fix $m = \mu = 0$, $\sigma^2 = 1$, and analyze the resulting posteriors for different sparsity coefficients $b$ (lower $b$ increases sparsity).

**Horseshoe.** The Horseshoe prior has become a popular probabilistic sparsity-inducing prior (Carvalho et al., 2009),

$$
\boldsymbol{\Lambda}_{ij}|\sigma, \tau \sim \mathcal{N}(0, \sigma^2\tau^2), \quad \sigma \sim C^+(0, 1)
\tag{26}
$$

where $C^+(0, 1)$ is a Half-Cauchy distribution for the local shrinkage $\sigma$, while $\tau$ is the global shrinkage parameter. The Horseshoe density of a single entry $\boldsymbol{\Lambda}_{ij}$ is

$$
\pi_\tau(\boldsymbol{\Lambda}_{ij}) = \frac{1}{\sqrt{2\pi^3\tau^2}} \exp\left(\frac{\boldsymbol{\Lambda}_{ij}^2}{2\tau^2}\right) E_1\left(\frac{\boldsymbol{\Lambda}_{ij}^2}{2\tau^2}\right),
\tag{27}
$$

where $E_1(\cdot)$ is the exponential integral function that can be approximated by elementary functions.

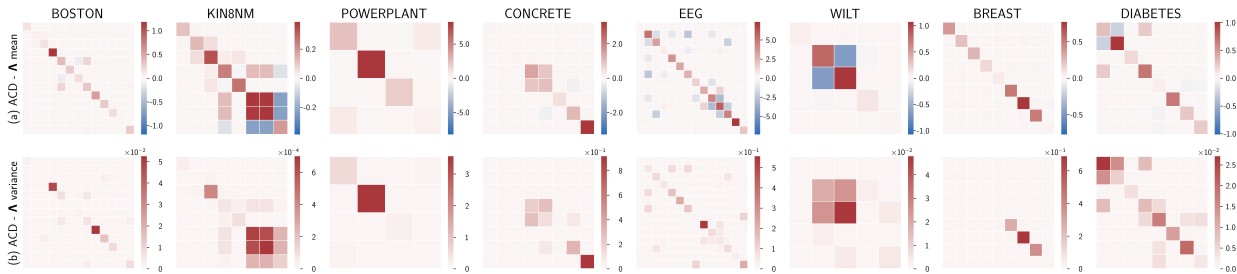

**Figure 3: The precision matrices reveal couplings, redundancies and separabilities.** The posterior mean (*a*) and variance (*b*) of precision matrices $\mathbf{\Lambda}$ of UCI benchmark datasets with Horseshoe prior ($\tau = 0.1$).

## 5 Experiments

We consider eight UCI datasets as a benchmark to assess performance of GP models for regression and classification tasks. We standardize all datasets to zero mean and unit variance, and report all results with five-fold cross-validation. Following previous works (e.g., Rossi et al. (2021)), we report test MNLL for all data, and normalized root mean square error (RMSE) for regression and error for classification tasks.

In all experiments, we chose to approximate GPs with 500 inducing points. We ran BSGP for 10,000 iterations with a step-size of 0.01 and mini-batch of 1,000 data points. We evaluate performance on test data from 50 samples collected during training after 1,500 burn-in iterations and using a thinning of 180. We adopt gradient clipping for numerical stability and to avoid exploding gradients, which we experienced when working with the Horseshoe priors.

### 5.1 UCI benchmarks

With the above setup, we report results on UCI benchmarks by considering various choices of priors (See Table 1). For the kernel variance $\sigma_f^2$ we placed a Lognormal prior with unit variance and mean 0.05 as in Rossi et al. (2021). The proposed MCMC scheme yields good convergence and sampling efficiency, as illustrated in Appendix C in the supplement; see also Fig. 19 and Fig. 20 for insights on the multimodality of the posterior. Fig. 2 shows the comparative performance for the UCI benchmark datasets, including the range between the 20th and 80th percentiles over the different folds, together with a rank summary. For the small data sets, we could also run full GPs and we observed a similar trend; we refer the reader to Fig. 15 and Fig. 16 for a direct comparison between full GPs and BSGPs.

Interestingly, different choices of prior and prior hyper-parameters yield comparable performance. A closer inspection indicates that the element-wise Laplace prior performs worst overall, and this might be due to the heavy sparsity promoted by this prior (or the lack thereof) for some hyper-parameter settings (Fig. 5). The element-wise Horseshoe prior, while promoting sparsity, fares slightly better than the Laplace prior. It is interesting how the inverse Wishart prior, which operates directly on $\mathbf{\Lambda}$, promotes some sparsity after all, while offering relatively competitive performance.

### 5.2 Sparse couplings

Next, we study the precision matrices themselves. Fig. 3 shows the posterior precisions of all benchmark datasets. Notably we see strong dependencies emerging in `kin8nm`, `eeg` and `wilt` datasets, while `powerplant`, `concrete`, `breast`, and `diabetes` are sparsely diagonal. We notice that the standard deviation of the elements on $\mathbf{\Lambda}$ is larger for covariate pairs with large positive/negative partial covariance, while it is generally small for pairs that have small partial covariance. This indicates both the relative scaling of uncertainty, and the flexibility in coupling magnitudes.

We provide a more in-depth look into the dependencies in Fig. 1 (page 2) that contrasts the precision matrix of the ARD covariance of `kin8nm` and `breast` datasets to the posterior mean and standard deviations of the

precision matrix of the ACD covariances. The ACD detects that 5th and 6th covariates of `kin8nm` are close to redundant, and negatively coupled to 7th covariate. Less evidently, in `breast` we detect coupling chains over covariates such as (0,3,6,8) and (1,6,7), indicating predictive dependencies in the data. Fig. 9 shows for this dataset how a different choice of the prior distribution over $\mathbf{\Lambda}$ can reveal a different and sparser structure of the couplings. We visualize these as circular graphs along with the standard deviations of the elements of the precision matrices.

Fig. 4 shows an ablation of comparing the posterior mean precision structures from Horseshoe prior with $\tau = \{0.01, 0.1, 1\}$ on the `concrete` dataset. The Horseshoe is able to sparsify the entire structure into an ARD-like structure, while higher $\tau = 1$ reveals off-diagonal dependencies. To obtain more intuition into the couplings, we also visualize the covariate graphs in the bottom panel of Fig. 4 that indicate for instance the strong dependence between the 3rd and the 4th covariate. Further illustrations on all UCI data sets for the Wishart prior Fig. 9, inverse Wishart Fig. 10, and Laplace prior with $b = 0.1$ Fig. 11 can be found in the supplement.

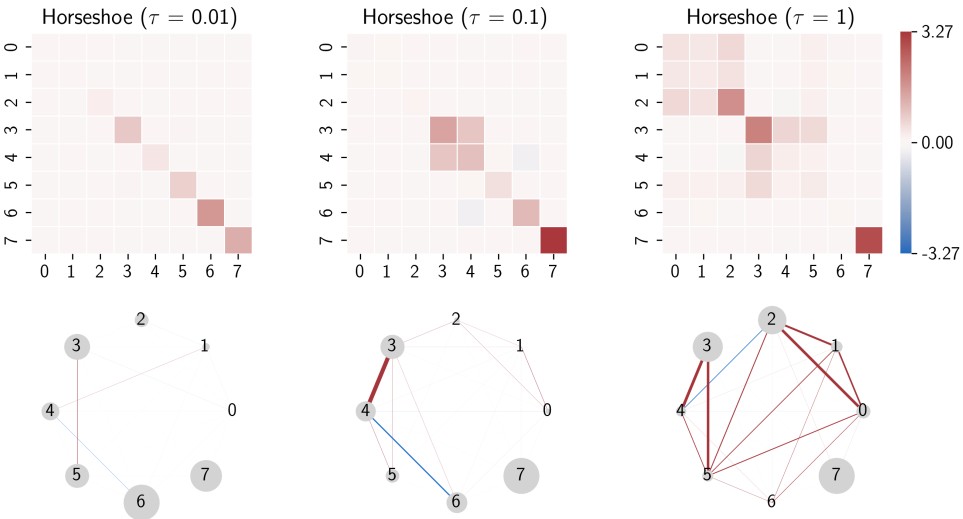

**Figure 4: The sparsity control of Horseshoe prior.** The posterior mean precision matrices of Horseshoe priors on `concrete` dataset with high (*left*) to low (*right*) sparsity.

### 5.3 Sparsification effect

Fig. 5 shows the sparsity of the posterior precision matrices $\mathbf{\Lambda}$ in the `boston` dataset. Surprisingly, the Inverse Wishart prior has an intrinsic sparsifying effect. The Laplace prior sparsifies according to its hyperparameter $b$ while, for this dataset, the Horseshoe prior with $\tau = 0.1$ achieves slightly more sparsity than the Horseshoe prior with $\tau = 0.01$.

As a conclusion of these experiments on UCI, we observe that the Horseshoe prior obtains better performance compared to the Laplace prior and it is competitive with matrix-variate priors. Also, these generally outperform the ARD covariance. Interestingly, there seems to be some data-dependent effect connecting sparsity and performance; in data sets such as `boston`, high sparsity seems to be associated with good performance, while for others such as `eeg` it is the opposite. This indicates that sparsity should perhaps be treated as a hyper-parameter and learned together with the model. We leave this interesting development for future work.

### 5.4 Low-rank precision matrices

We also look at the effect of low-rank precision matrices. Fig. 6 shows the posterior precision patterns learned by the Wishart prior using a PCA with rank 11, 7 or 3 in contrast to the full rank 13. The performance

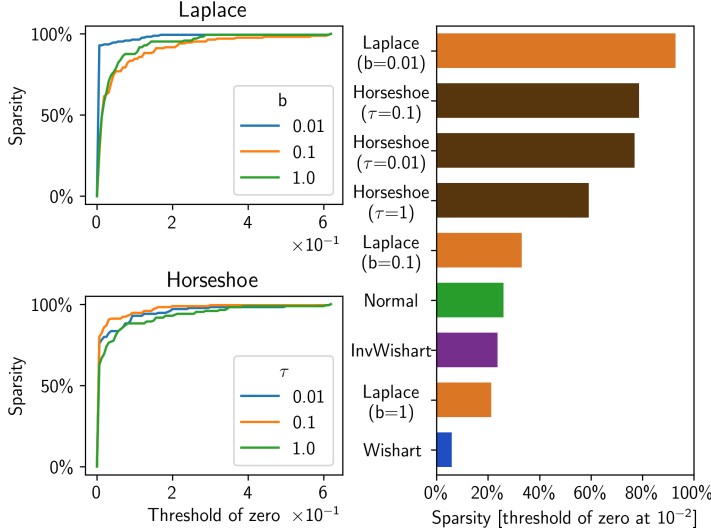

**Figure 5: The sparsification of `boston` dataset**. Left: Relationship between precision sparsity and hyperparameters. Right: Posterior mean sparsity from different priors.

degrades strongly at ranks lower than 11, which is likely indicative of the intrinsic rank of the dataset for this task.

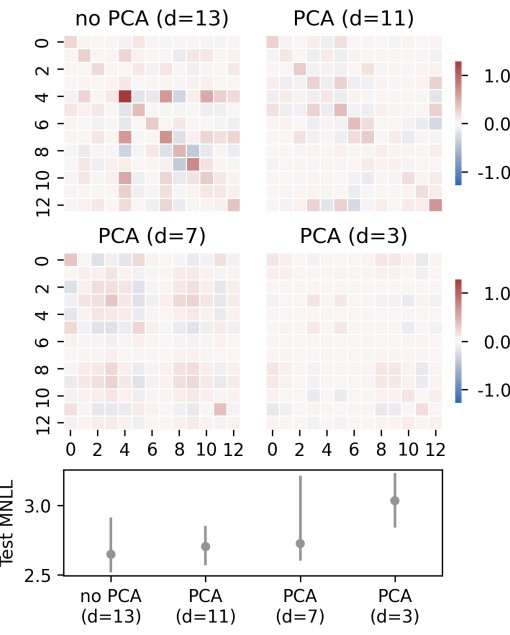

**Figure 6: Overly low rank degrades performance.** Posterior precisions with Wishart prior of varying rank $d$ of Eq. (14) (*top*) and corresponding performances (*bottom*) on the `boston` dataset.

## 5.5 Dependencies of motion capture data

We illustrate the capability of ACD covariances to reveal dependencies in a motion capture task, where the subjects internal connectivity is known (Fig. 7). We observe a trajectory $\mathbf{Y} = (\mathbf{y}_1, \mathbf{y}_2, ... \mathbf{y}_N)^T \in \mathbb{R}^{N \times D}$ over

**Table 2:** MoCap results on subject 09 using GP-ODE with ARD and ACD kernels.

| Metric | Method | Subject 09 (short) |
|---|---|---|
| MNLL | GP-ODE-vanilla ARD | $-2.25 \pm 0.18$ |
|      | GP-ODE-vanilla ACD | $-2.13 \pm 0.17$ |
| MSE  | GP-ODE-vanilla ARD | $34.49 \pm 5.60$ |
|      | GP-ODE-vanilla ACD | $51.17 \pm 12.97$ |

$N$ timepoints, where $\mathbf{y}_i \in \mathbb{R}^D$ represents the noisy observation of subject state $\mathbf{x}(t_i) \in \mathbb{R}^D$ at time $t_i$. The state consists of a total of $D = 50$ measurements across 21 body parts (Fig. 7). We follow the GP-ODE model (Heinonen et al., 2018; Hegde et al., 2022), where the state follows an ordinary differential equation $\dot{\mathbf{x}}(t) = \mathbf{f}(\mathbf{x}(t))$ with a vector-valued GP prior on the differential $\mathbf{f} : \mathbb{R}^D \mapsto \mathbb{R}^D$,

$$\mathbf{f(x)} \sim \mathcal{GP}(\mathbf{0}, K_{\boldsymbol{\theta}}(\mathbf{x}, \mathbf{x}')), \tag{28}$$

where $K_{\boldsymbol{\theta}} \in \mathbb{R}^{D \times D}$ is an operator-valued kernel. The most straightforward covariance function is a separable one $K(\mathbf{x}, \mathbf{x}'; \boldsymbol{\theta}) = k(\mathbf{x}, \mathbf{x}'; \boldsymbol{\theta})\mathbf{I}_D$, where we learn a shared precision matrix for all outputs. As an alternative, we also consider a variant $K(\mathbf{x}, \mathbf{x}'; \boldsymbol{\theta}) = \text{diag}\{k(\mathbf{x}, \mathbf{x}'; \boldsymbol{\theta}_1), \ldots, k(\mathbf{z}, \mathbf{z}'; \boldsymbol{\theta}_D)\}$, where each diagonal entry has its own kernel $k(\mathbf{z}, \mathbf{z}'; \boldsymbol{\theta}_i)$ and its own precision matrix $\boldsymbol{\Lambda}_i$ associated with output $\dot{x}_i$.

Fig. 7 shows the posterior shared precision mean pooled over the body parts in a human walk cycle. A rich pattern of dependencies emerges. For instance, right and left wrists are strongly coupled across the body, while being negatively coupled to each other. The wrists move in large, cyclic and synchronised patterns, while the back and root have little relevance, indicating their smaller movement ranges during walking. Finally, many adjacent body parts are coupled, such as foot and tibia, and wrist and radius. Table 2 shows the performance between ARD and ACD on subject 09, where the likelihoods are similar, but ACD does perform worse in mean square error. The purpose of the experiment was to demonstrate structure learning with standard inference runs, and we did not focus on performance tuning, which ODE models are known to be finicky about (Hegde et al., 2022).

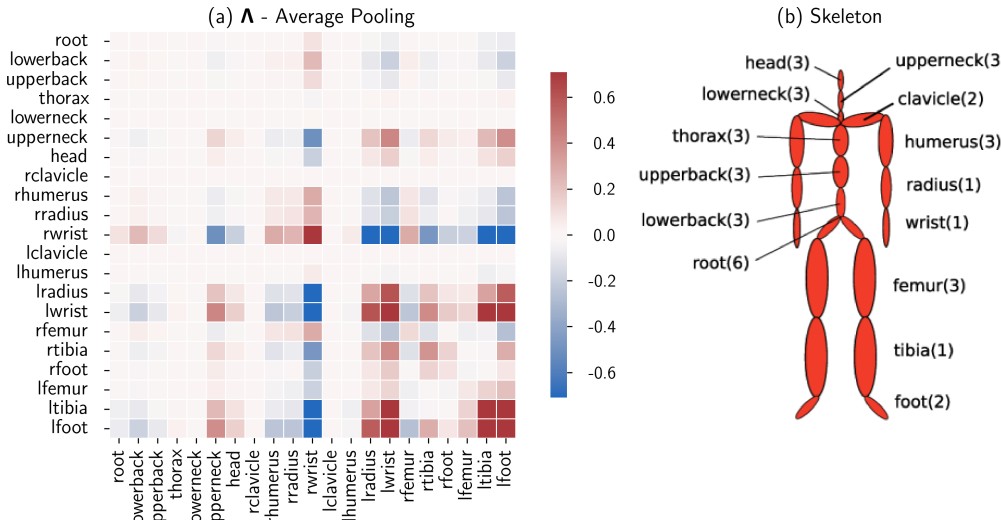

**Figure 7: The ACD covariance reveals a highly regular coupling structure from human motion.** GP-ODE model trained with shared ACD covariance in a latent space of 15 dimensions. Panel (*a*) shows the precision matrix $\boldsymbol{\Lambda}$ reporting just the average value for each group of sensors. Panel (*b*) shows the reference skeleton connectivity.

## 6    Conclusions

In the literature of GPs, covariances equipped with ARD are popular. These materialize with the definition of a set of length-scale parameters scaling the inputs, which are then optimized or inferred based on the marginal likelihood (or an approximation/bound). In this work, we revisited a more general definition of anisotropic covariances, where the distance metric among inputs is determined by a PSD precision matrix. We showed that this extension provides a framework for metric learning and we discussed some interesting insights on the determination of couplings among covariates. Crucially, thanks to a fully Bayesian scalable formulation of GPs, we can operate with virtually any number of data points and obtain samples from the posterior distribution over such covariate couplings, which can be used to determine the level of confidence in their predictive power.

We also studied priors for the precision matrix $\mathbf{\Lambda}$ determining the input metric. We showed that element-wise Laplace and Horseshoe priors provide the highest level of sparsity, while Horseshoe priors seem to offer better performance. Interestingly, the inverse Wishart prior offers higher sparsity than the Wishart prior with overall comparable performance.

In order to address the quadratic scalability with respect to the number of covariates, we also revisited the work by Vivarelli & Williams (1998), which proposes a low-dimensional projection of the inputs through PCA, in light of modern scalable GPs and inference.

We are currently investigating an extension of our approach whereby the conclusions we can draw from the analysis of $\mathbf{\Lambda}$ are in terms of conditional independence statements. In order to do this, we plan to extend our model to target the modeling of both labels and inputs, including a prior over the inputs $p(\{\mathbf{x}_n\}|\mathbf{\Lambda})$ in the form of a Markov Random Field, where $\mathbf{\Lambda}$ now determines the conditional independence among covariates.

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

# A    Experimental details

In this section, we present details to reproduce our experimental campaign. All the experiments were conducted on Google Colab.

**BSGP model**    We use $M = 500$ inducing points initialized by a k-means algorithm as commonly used in practice and we place a Normal prior $p_{\boldsymbol{\xi}}(\mathbf{Z})$ over the inducing locations $\mathbf{Z}$. For inference, we use an adaptive version of Stochastic Gradient Hamiltonian Monte Carlo (SGHMC) in which the hyperparameters are automatically tuned during a burn-in phase. We set the default hyperparameter of the number of SGHMC steps to $K = 10$. Exclusively for regression datasets with Gaussian likelihood, we employ an Adam optimizer with a learning rate set at 0.01 for optimizing the variance of the likelihood.

**ARD kernel**    We use the  Radial Basis Function (RBF) kernel with  Automatic Relevance Determination (ARD) placing a LogNormal prior with unit variance and means equal to 1 and 0.05 for the lengthscales and variance, respectively.

**ACD kernel**    We place a LogNormal prior with unit variance and mean 0.05 over the kernel variance $\sigma_f^2$ while over the precision matrix $\boldsymbol{\Lambda}$ we explore a wide range of priors.

**Table 3:** Parameter settings for the UCI experiments.

| parameter | value |
|---|---|
| num. of inducing points | 500 |
| mini-batch size | 1000 |
| num. iterations | 10500 |
| step size | 0.01 |
| momentum | 0.05 |
| num. of burn-in steps | 1500 |
| num. of samples | 50 |
| thinning interval | 180 |

# B    Simulated dataset

In this section we carry out an experiment using simulated datasets with known underlying precision matrices. In particular we assess the ability of the BSGP model using a ACD kernel to recover the true precision $\boldsymbol{\Lambda}$ while fitting simple regression problems. Here it's described how the simulated regression datasets are constructed and some experiments conduced to show the behaviour of the model. We consider $N$ input-label pairs $\{\mathbf{X}, \mathbf{y}\} = \{(\mathbf{x}_n, y_n)\}_{n=1}^N$ with $\mathbf{x}_n \in \mathbb{R}^D$ and $y_n \in \mathbb{R}$ defined as follows:

$$
\begin{aligned}
\mathbf{x}_n &\sim \mathcal{N}(\mathbf{0}, \mathbf{I}) \\
\mathbf{K}_{\mathrm{xx}} : \mathbf{K}_{\mathrm{xx}}[i,j] &= \sigma_f^2 \left( (\mathbf{x}_i - \mathbf{x}_j)^\top \boldsymbol{\Lambda} (\mathbf{x}_i - \mathbf{x}_j) \right) \\
\mathbf{y} &\sim \mathcal{N}(\mathbf{0}, \mathbf{K}_{\mathrm{xx}} + \sigma_n \mathbf{I})
\end{aligned}
\tag{29}
$$

Once that the underlying precision $\boldsymbol{\Lambda}$ has been constructed, specifying a value for the kernel variance $\sigma_f^2$ and another one for the Gaussian noise in observations via $\sigma_n$ is sufficient. The regression dataset $\{\mathbf{X}, \mathbf{y}\}$ can be used to train a BSGP model by means of the ACD kernel. Through acquiring samples of the precision matrix $\boldsymbol{\Lambda}$, we aim to recover the original underlying precision used to generate the data. A visual insight into this experiment is given in Fig. 8.

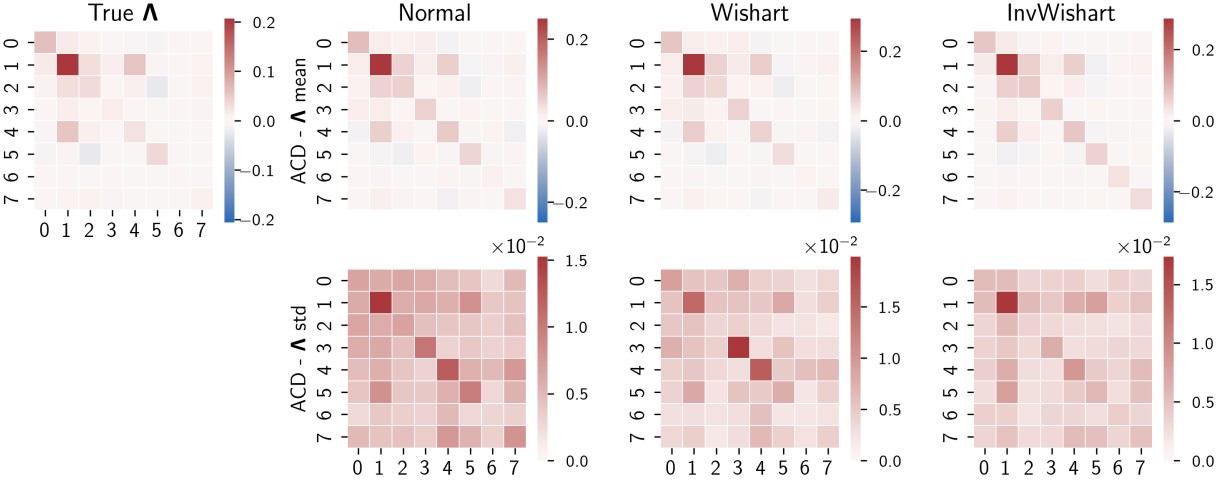

**Figure 8:** Underlying sparse precision (top left) compared with mean and standard deviation of the $\mathbf{\Lambda}$ samples obtained with different priors. The dataset is made of $N = 1000$ samples and the labels are obtained according to Eq. 29 setting $\sigma_f^2 = 1$ and $\sigma_n = 0.1$

.

## C  Additional results

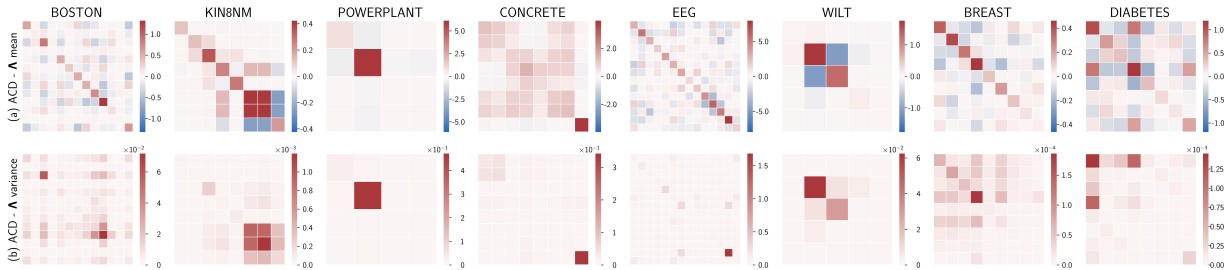

**Figure 9:** Posterior precision matrix mean ($a$) and variance ($b$) with Wishart prior.

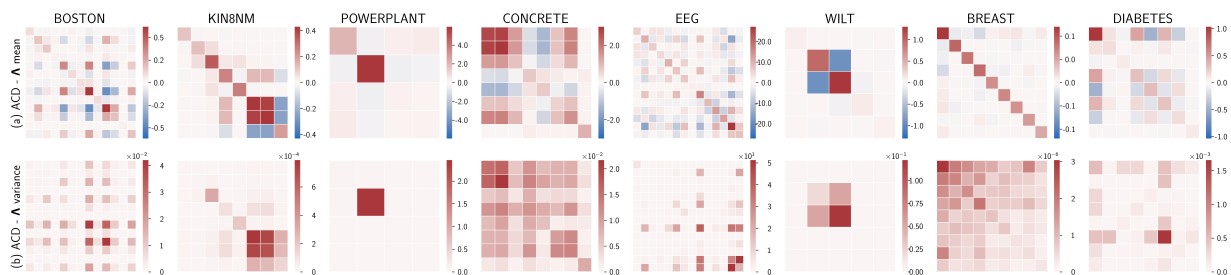

**Figure 10:** Posterior precision matrix mean ($a$) and variance ($b$) with Inverse Wishart prior.

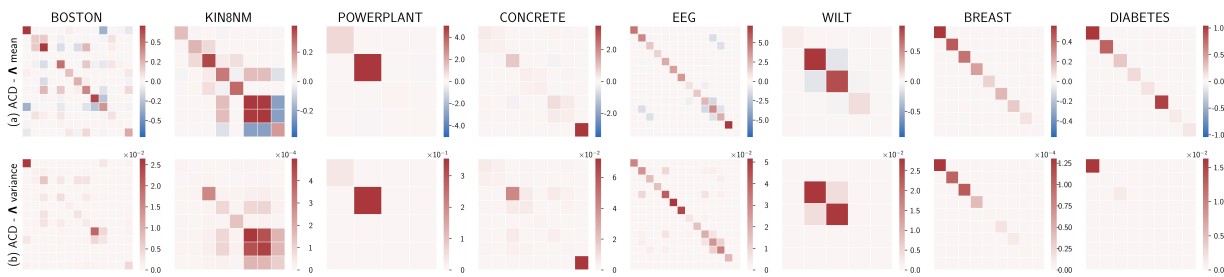

**Figure 11:** Posterior precision matrix mean ($a$) and variance ($b$) with Laplace prior $b = 0.1$.

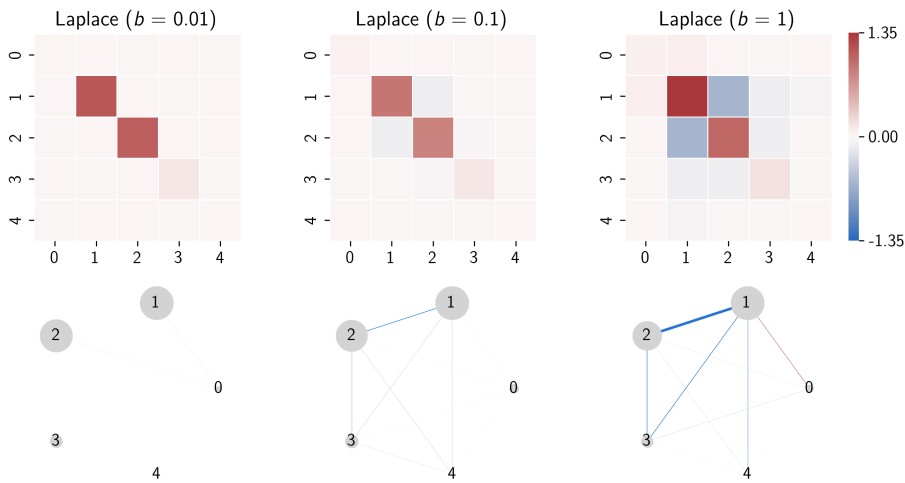

**Figure 12:** The precision matrices of the `wilt` dataset using Laplace prior show a progressive level sparsity.

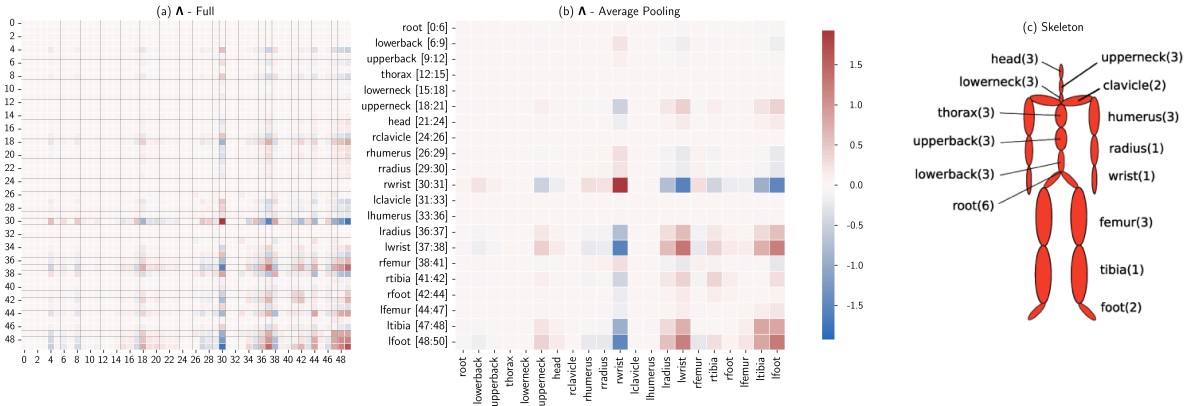

**Figure 13:** The full motion capture precision matrix ($a$), a pooled part-wise ($b$) and reference skeleton connectivity ($c$).

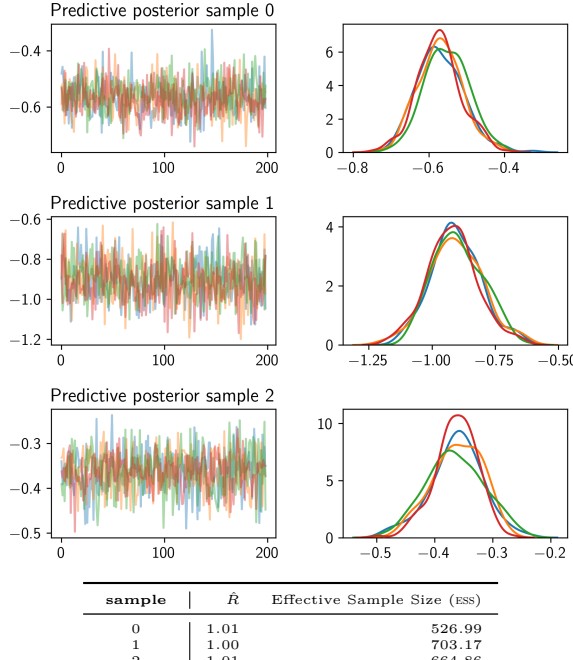

| sample | $\hat{R}$ | Effective Sample Size (ESS) |
|--------|-----------|------------------------------|
| 0 | 1.01 | 526.99 |
| 1 | 1.00 | 703.17 |
| 2 | 1.01 | 664.86 |

**Figure 14:** Traces of the mean of the predictive distribution for three test points on `boston` dataset with Inverse Wishart prior (4 chains, 200 samples represented); the table reports $\hat{R}$ and Effective Sample Size (ESS) statistics for each set of 4 chains.

**Table 4:** UCI datasets used, including number of datapoints and dimensionalities.

| Dataset | $N$ | $D$ |
|---------|-----|-----|
| boston | 506 | 13 |
| breast | 683 | 9 |
| diabetes | 783 | 8 |
| concrete | 1,030 | 8 |
| wilt | 4,839 | 5 |
| kin8nm | 8,192 | 8 |
| powerplant | 9,568 | 4 |
| eeg | 45,730 | 14 |

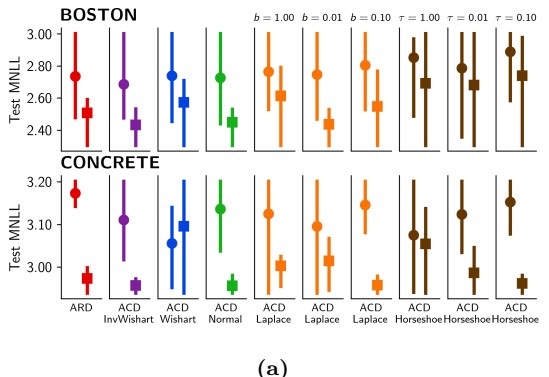

(a)

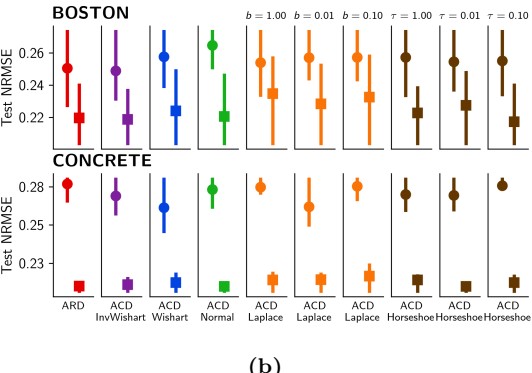

(b)

**Figure 15:** Comparison of full GPs (□) vs BSGPs (○) with 200 inducing points on two UCI regression data sets. The metrics are MNLL in **(a)** and normalized RMSE in **(b)**.

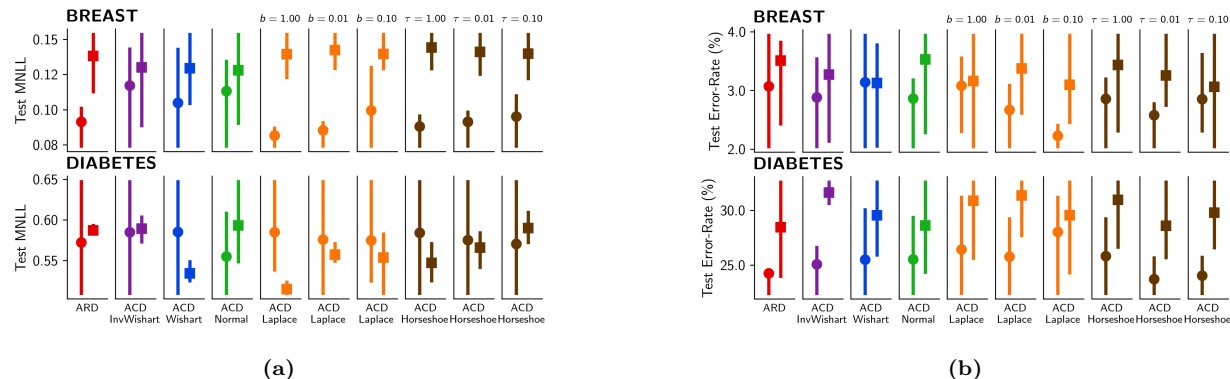

**Figure 16:** Comparison of full GPs (□) vs BSGPs (○) with 500 inducing points on two UCI classification data sets. The metrics are MNLL in **(a)** and Error-Rate in **(b)**.

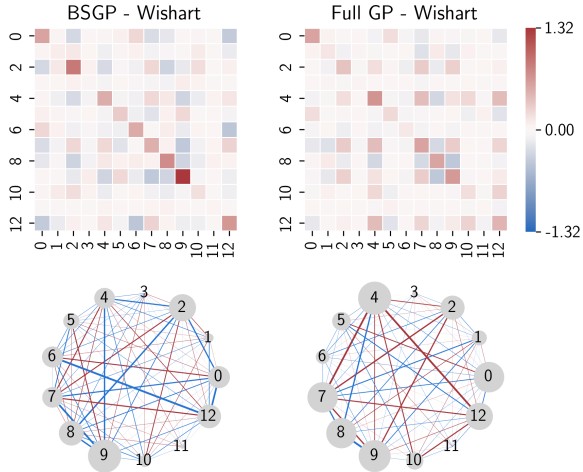

**Figure 17:** Comparison of posterior mean of the precision matrix **Λ** on the `boston` dataset with Wishart prior for full GP vs BSGP with 500 inducing points.

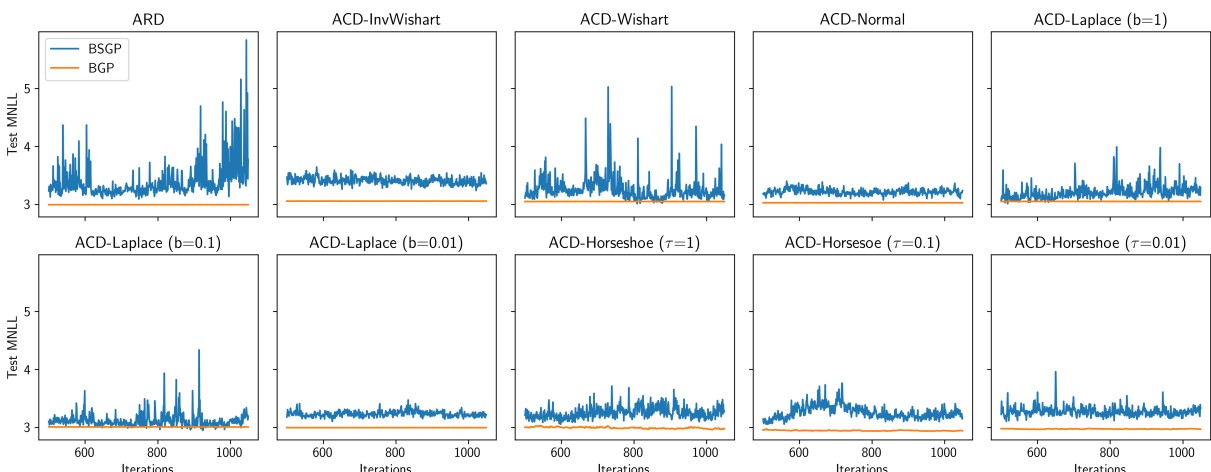

**Figure 18:** MNLL vs iterations for BSGP with 500 inducing points and for full GP on `concrete` dataset. The plots show one value every 10 of the 10,000 iterations.

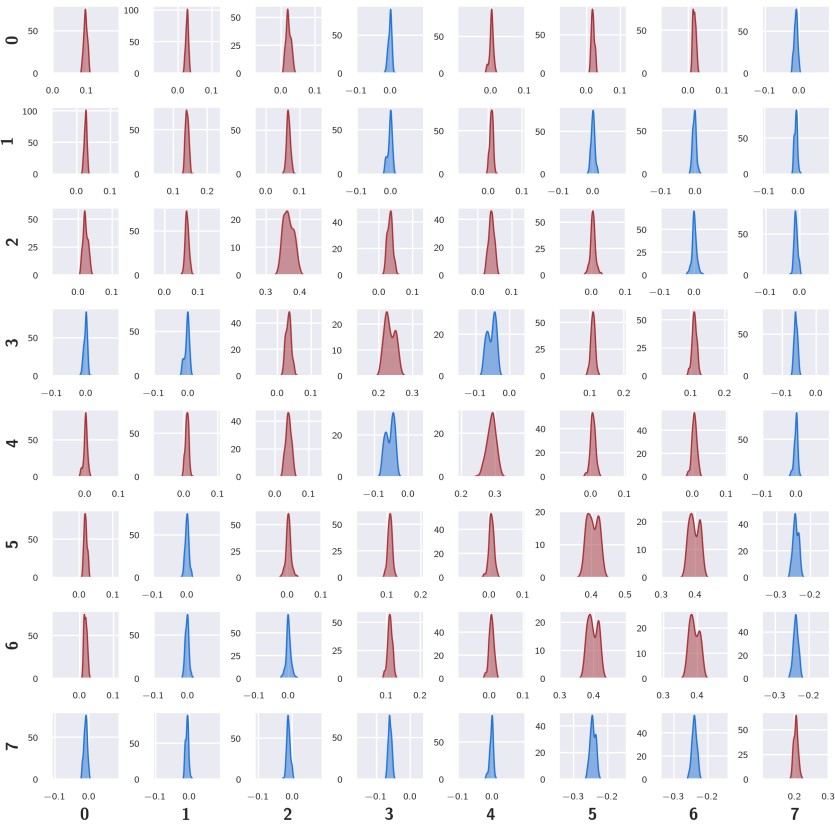

**Figure 19:** Posterior samples distribution of precision matrix entries for `kin8nm` dataset with Horseshoe ($\tau = 0.1$) prior.

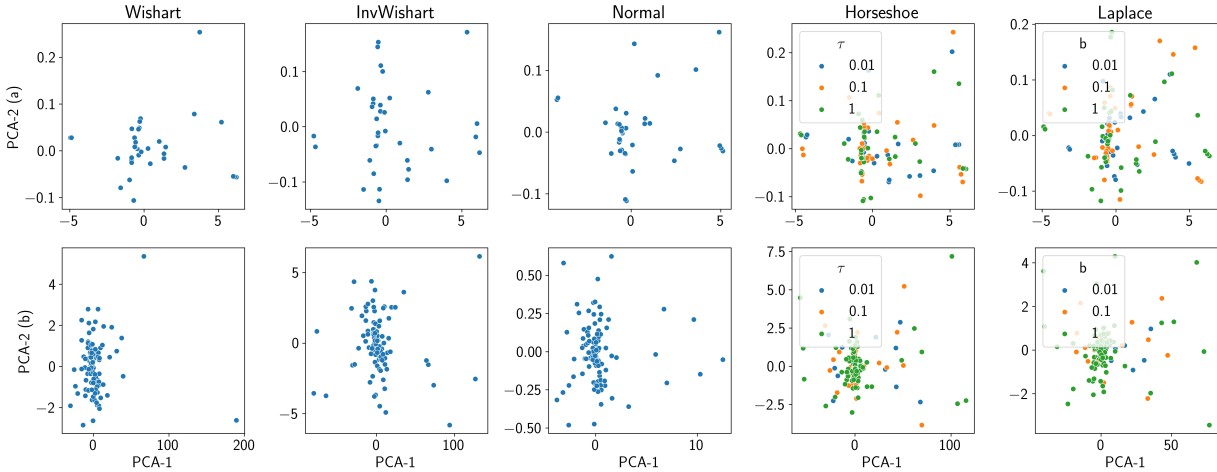

**Figure 20:** PCA representation of vectorized posterior precision matrices. Each point in the 2D space represents a posterior sample (precision matrix). *(a)* `kin8nm` dataset, *(b)* `eeg` dataset.

