# OpenReview forum: "Gaussian Processes with Bayesian Inference of Covariate Couplings"
_TMLR — Rejected by TMLR_

### Review · Reviewer_5ys6 · 2024-09-26

**Summary Of Contributions:**

This paper studies the following problem.
Suppose we have a zero mean Gaussian process $GP(0,k)$ where $k$ is a positive definite kernel function.
In particular, this paper consider the kernel function of the form $\exp(-\frac{1}{2}d^2(x,y))$ where $d$ is a distance function.
The simplest form of $d$ is $\|x-y\|^2$.
The Gaussian process of the above form is called Automatic Relevance Determination.
However, the drawback of this form is it does not capture the dependence between variables.
In order to address this issue, the authors propose a form by considering $d$ to be $(x-y)^\top \Lambda (x-y)$ for some positive definite matrix $\Lambda$.

**Audience:**

Yes

**Claims And Evidence:**

Yes

**Requested Changes:**

- First paragraph in Section 2:
Is the subscript in $\prod_{i=1}^N p(y_n \mid f_n)$ $i$?
Is $f_n$ $f(\mathbf{x}_n)$?

**Strengths And Weaknesses:**

Strengths:

- The authors introduced the concept of Automatic Coupling Determination (ACD), which enables the consideration of dependencies between variables.
This is a promising direction to pursue, as different features in real-world data are often interdependent.



Weaknesses:

- This paper focuses on the empirical evaluation of the proposed method.
However, it lacks theoretical guarantees regarding performance metrics such as convergence rates and sample complexity.
This omission may make it challenging for readers to fully assess the strengths of the proposed method.

---

> ### Author Response · Authors · 2025-02-12
> **Q: The authors introduced the concept of Automatic Coupling Determination (ACD), which enables the consideration of dependencies between variables. This is a promising direction to pursue, as different features in real-world data are often interdependent.**
>
> We thank the reviewer for the positive feedback acknowledging the value of our contribution.

---

> ### Author Response · Authors · 2025-02-12
> **Q: This paper focuses on the empirical evaluation of the proposed method. However, it lacks theoretical guarantees regarding performance metrics such as convergence rates and sample complexity. This omission may make it challenging for readers to fully assess the strengths of the proposed method.**
>
> Many thanks for this comment, which allows us to expand on the computational aspects of our proposal. We created Section 4.4 on computational complexity in the revision. This sub-section opens with comments on the sparse GP approach, for which we have complexity in $O(M^3)$ to factorize the matrix $K_{uu}$ and $O(N M^2)$ to multiply $K_{fu}$ with the inverse (or do forward/backward substitution with the Cholesky of) $K_{uu}$. These operations are needed to compute the predictive distribution. Within the MCMC sampling approach in Rossi et al., (2021), mini-batching allows to reduce the complexity of $O(N M^2)$ to $O(N’ M^2)$ with $N’ << N$
>
> In the second part of this sub-section, we comment on the fact that evaluating each element of $K_{uu}$ and $K_{uf}$ requires $O(D^2)$ computations for ACD covariance functions, and this is the case when evaluating element-wise priors and matrix variate priors on $\Lambda$. For ARD covariances, instead, this cost is $O(D)$. To be more specific, for $K_{uu}$ the complexity is $O(M^2D^2)$, since $K_{uu}$ is $M \times M$ and each of the $M^2$ entry involves matrix-vector products with $\Lambda$ matrix. For the $M \times N$ matrix $K_{uf}$ it is $O(MND^2)$.

---

> ### Author Response · Authors · 2025-02-12
> **Q: First paragraph in Section 2: Is the subscript in the likelihood i? Is f_n = f(x_n)**
>
> Thanks for spotting this typo, which we fixed in the revision.

---

### Review · Reviewer_3bqA · 2024-11-11

**Summary Of Contributions:**

This paper examines a class of nonlinear regression problems using Gaussian process approaches. The covariance functions of the Gaussian process are modeled with Gaussian kernels, with the covariance matrix treated as a parameter for inference within a Bayesian framework. The paper explores and discusses different priors for the covariance matrix in the inference process.

**Audience:**

No

**Broader Impact Concerns:**

This paper examines theoretical topics and has no direct ethical implications.

**Claims And Evidence:**

Yes

**Requested Changes:**

1. Please clarify the relation between y and f(x). We know that usually in nonlinear non-parametric regression models, there is noise assumed, y=f(x)+ε. We see this model in the current manuscript, e.g., in Equation (2), as p(y|f). However, we also see that the vector f(x), given x, is by itself also random. This is seen in Equation (1) as that f(x) has a gaussian process, and also in Equation (4) that although u=f(z), the probability of u given Z, p(u|Z, θ), is still not degenerate. We would like the authors to clarify why such double-randomness setting is necessary.

2. What is "f" in (3), (4), and (5)? Is this f the same as, or different from the "f" in (1) and (2)?

**Strengths And Weaknesses:**

weaknesses
1. I believe this paper lacks a sufficient description of the background and motivations. The connection between the application context and the approaches adopted is unclear.
2. The boundary between the literature, and the contributions provided in this paper, is unclear.

---

> ### Author Response · Authors · 2025-02-12
> **Q: I believe this paper lacks a sufficient description of the background and motivations.**
>
> We thank the reviewer for this comment, which allows us to clarify the background and motivations. We consider supervised learning problems and we ask ourselves the question of whether we can develop models which are both flexible and, to some extent, interpretable. Gaussian process models possess these characteristics when covariance functions are chosen with this goal in mind. However, to date, there has been little work beyond covariance functions which weigh features globally (isotropic covariances) or individually (for Automatic Relevance Determination (ARD)).
>
> In this work, we study more expressive covariance functions which perform affine transformations of the features through a matrix $\Lambda$, thus allowing us to uncover couplings among them; for this reason we term these Automatic Coupling Determination (ACD) covariances. While these covariance functions were proposed before (Vivarelli & Williams, 1998), we are not aware of studies which treat $\Lambda$ in a Bayesian manner apart from the work in Titsias & Lazaro-Gredilla (2013) who carried out mean-field variational inference on $\Lambda$. Also, we are not aware of previous studies on the effect of sparsity-induced priors on $\Lambda$ and the treatment of these through scalable MCMC sampling. For these reasons, we believe that our contributions are valuable in the context of increasing the flexibility and expressiveness of Gaussian process models, while operating in a scalable way both for the modeling (sparse Gaussian processes) and in the inference (scalable MCMC).
>
> We emphasized these points in the introduction.

---

> ### Author Response · Authors · 2025-02-12
> **Q: The connection between the application context and the approaches adopted is unclear.**
>
> In several applications the features exhibit some level of redundancy and it can be useful to identify such cases so as to obtain more parsimonious representations of the data or to inform the design of future experiments. The approach we propose here is a step in the direction of revealing such couplings, while performing a nonparametric supervised learning task. We tried to offer insights on these observations with a number of illustrative toy examples and with applications in human motion, where we can match the findings from the inference over ACD covariance parameters with our intuition on the dependency structure associated with this application.
>
> We emphasized these points in the revision of the paper in the introduction and in Section 6.5.

---

> ### Author Response · Authors · 2025-02-12
> **Q: The boundary between the literature, and the contributions provided in this paper, is unclear.**
>
> Many thanks for this comment, which allows us to emphasize what parts of our work are novel and to better position our work within the literature. The background section (Section 2) and the beginning of Section 3 on covariance functions are relatively standard background material, while the majority of Section 3 is partially novel and Section 4 contains novel developments. Here is a short summary of the related literature and the positioning of our work.
>
> In the revision, we added the following text in a new section on “Related Works”.
>
> **Gaussian processes with Automatic Coupling Determination**
>
> The possibility to carry out kernel-based modeling with a determination of the importance of inputs dates back at least to the works on Automatic Relevance Determination (ARD) (MacKay, 1995; Neal, 1996). This is usually implemented by scaling input covariates within the calculation of the covariance function by some coefficients which are treated as hyper-parameters and optimized through marginal likelihood optimization.
>
> An extension of this idea involves the use of affine transformations (rotation and stretching) of the covariates; in distance-based covariance functions, the affine transformation implies the calculation of the so-called Mahalanobis distance (Vivarelli & Williams, 1998; Titsias & Lazaro-Gredilla, 2013). In Vivarelli & Williams (1998), $\Lambda$ is made positive definite by construction through the parameterization $U U^{\top}$ with $U$ upper triangular, and it is factorized to gain insights into the dimensionality of a possible low-dimensional latent representation of the inputs. In our work, we consider a similar parameterization for $\Lambda$, but instead of optimizing its factors, we carry out a Bayesian treatment, for which we study sparsity-inducing priors. Also, we propose a PCA-based decomposition of $\Lambda$, which allows us to operate in large-dimensional input regimes; this is done through the first $d$ principal components $P_d$ of the input covariance, which we use to express $\Lambda = P_d \Lambda_d P_d$.
>
> The work by Titsias & Lazaro-Gredilla (2013) considers the parameterization $\Lambda = W^{\top} W$ without imposing any structure on $W$ except for imposing that $W$ maps the input covariates to a lower dimensional input space. Their work proposes a variational formulation to obtain an approximation to the posterior over $\Lambda$ but it does not extensively study priors over $W$, which is the focus of our work.
>
> **Sparsity-Inducing priors for covariance/precision matrices**
>
> The literature on Gaussian Graphical Models (GGMs) provides studies on sparsity-inducing priors for covariance and precision matrices. Sparsity can be imposed in a structured fashion by considering graph decomposability (Banerjee and Ghosal, 2014; Lee and Lee, 2021; Xiang et al., 2015, Banerjee et al., 2021), or through the G-Wishart prior, which has been introduced as a conjugate prior for the precision matrix in a Gaussian framework and it is also suitable for cases where graph decomposability does not apply (Roverato, 2000; Roverato, 2002; Khare and Rajaratnam, 2011; Liu and Martin, 2019; Silva and Ghahramani, 2009; van den Boom et al., 2022).
>
> Various approaches have been developed to carry out inference in GGMs, including Gibbs sampling (Khare and Rajaratnam, 2011; Wang, 2012) and Laplace approximations (Banerjee and Ghosal, 2015). Other approaches, such as those by Gan et al. (2019) and Wang (2015), propose spike-and-slab sparsity-inducing priors which typically complicate posterior sampling. Castillo et al., (2015), Li et al., (2019), and Sagar et al., (2024) study horseshoe priors, which perform well in practice.
>
> Our work differs from this literature, given that we propose a model for the labels given the inputs, while attempting to uncover some couplings among covariates. The way this is done is by a parameterization of the covariance function akin to the precision matrix in a GGM, and in our work we explore both matrix-variate and element-wise priors for such model parameters. In addition, we consider scalable sampling-based approaches to obtain samples from the posterior distribution over these parameters.

---

> ### Author Response · Authors · 2025-02-12
> **Q: Please clarify the relation between y and f(x) [...]**
>
> Thanks for the comment. In Gaussian processes we aim at characterizing the posterior distribution over a random function $\phi$, for which we assume a GP prior, after processing some data in the form of $N$ inputs $X$ and associated labels $y$. This is numerically costly with cubic scaling with respect to the number of input points $N$. Sparse GPs alleviate this issue by introducing the set of auxiliary (pseudo-data) variables $u$, of size $M<N$, which represents the realization of the random function $\phi$ at inputs $Z$. Denoting by $f$ the realization of $\phi$ at $X$, our new goal is to approximate the joint posterior $p(f,u|y)$ over the two unknowns $f,u$. We now have two sets of latent random variables, but the source of randomness is still encoded in the process $\phi$, so by augmenting our model by considering the realization of the process at some new input locations, we are not adding any extra sources of randomness. Sparse GPs leverage the properties of multivariate Gaussian distributions to obtain a formulation which scales linearly with $N$ and cubically with $M$, which is advantageous when $M << N$.
>
> In the revision we spent a few more words on the construction of the model, by clarifying what the likelihood function is in Section 2.1 in the cases of regression and classification (which we consider here), and by clarifying the sparse Gaussian process construction in Section 2.2.

---

> ### Author Response · Authors · 2025-02-12
> **Q: What is "f" in (3), (4), and (5)? Is this f the same as, or different from the "f" in (1) and (2)?**
>
> Thanks for this question, which allows us to clarify this point which was imprecise in the original submission. The random variables $f$ which appear in equations (2,3,5) are the realization of the random function, defined in equation (1), at the inputs $X$. In the sparse model in Section 2.2, “auxiliary” variables $u$ are introduced in order to derive a scalable inference framework (e.g., Titsias, AISTATS 2009). In practice, the sparse GP equations (3-5) lead to somewhat different estimates of the function $f$ than in standard GP equations (1-2).
>
> We have modified the text in Section 2.1 to fix this inconsistency.

---

### Review · Reviewer_zGky · 2024-12-17

**Summary Of Contributions:**

The authors investigate extracting couplings between input features by training Gaussian Processes with adjustable whitening of the inputs. They provide simulations carried out with fully Bayesian inference.

**Audience:**

No

**Broader Impact Concerns:**

I have no concerns about the broader impact.

**Claims And Evidence:**

Yes

**Requested Changes:**

# Required
- [C1] Provide analytical or empirical evidence that ACD GPs are more effective in identifying couplings than classical data analysis (PCR).
- [C2] Related to C1: Carve out the differences and advantages between ACD and PCA+ARD.
- [C3] Provide computation times.

# Questions
- [Q1] Features obtained from PCA immediately justify ARD because they are mutually orthogonal. Why should we need full-precision matrices?
- [Q2] Cf. Section 5.4. Are the non-diagonal couplings in lower-rank ACD matrices just projection errors incurred by cutting the rank of the PCA, thus leaving some variance unexplained?

**Strengths And Weaknesses:**

TL;DR
Identifying covariate couplings is important for interpretability, but the approach taken by the authors does not convince me.

# Strengths
- [S1] The problem of identifying intrinsic structure in the data is interesting.
- [S2] The fully Bayesian treatment is nicely motivated and explained.
# Weaknesses
- [W1] The paper reads like a tutorial-style paper.
- [W2] What is the cost of using full precision matrices? The paper does not provide computation times. Do ARD models get way less compute?
- [W3] Why infer covariate couplings simultaneously with regression? As mentioned by the authors, a standard method to work with high-dimensional data is to project it on their principle components via Principle Component Analysis. PCA extracts the whitening without costly Bayesian inference.
- [W4] The experimental section has no proper hypotheses. For example: "ACD with prior x outperforms ARD." or "ACD with prior x induces higher sparsity levels than ARD." with corresponding evidence/ statistical analysis. Section 5.1 reports observations in a handwavy fashion.
- [W5] The experiments leave the question of whether sparsity is desired unanswered. The future work on statements about conditional independence of inputs seems relevant here.

---

> ### Author Response · Authors · 2025-02-12
> **W1: The paper reads like a tutorial-style paper**
>
> Thank you for this comment. We tried to carefully balance background exposition with the presentation of new technical developments. We aimed at producing a self-contained presentation, since our paper revisits ideas that originate in the 90’s and develops them in light of recent advances in MCMC for GPs and sparsity-inducing priors for matrix-variate variables; we hope that the current presentation style gives readers all necessary background to appreciate the developments in the paper. The new changes should help readers to identify more clearly the main contributions of our paper.

---

> ### Author Response · Authors · 2025-02-12
> **W2:  What is the cost of using full precision matrices? The paper does not provide computation times. Do ARD models get way less compute?**
>
> Thank you for this comment, which encouraged us to write more explicitly about computational complexity when employing ACD covariances and contrast them with ARD covariances (see new Section 4.4). The ACD covariance requires the multiplication of the inputs by a $D \times D$ matrix (cost in $O(D^2)$, while in ARD this cost is linear due to the multiplication with a diagonal matrix. Notably, we don’t need any matrix inverses here since we work with a precision matrix. We expect this cost to be significant for very large $D$, but Gaussian processes are often applied to tabular data, where the number of covariates $D$ is limited, so in practice there is little runtime difference. In the case of large $D$, we proposed an alternative low-rank parameterization of $\Lambda$ which scales favorably with $D$ (equation (14)) given that principal components can be found without a full decomposition of the $D \times D$ input covariance matrix.
>
> In addition to the complexity analysis above, we have also added one extra experiment showing the wall clock difference between running sparse GPs with the ARD and ACD covariance. The setup is similar to the one used to obtain Figure 2, except that we used 200 inducing points instead of 500. We ran the MCMC sampler for 10500 training iterations (1500 burn-in steps and we collected one samples every 180, yielding a total of 50 samples) on a server with A100 GPUs. Results are averaged across three folds. The running times are on the same order of magnitude, and the model with the ARD covariance is about 20% faster than the model with the ACD covariance.
>
> **Runtime comparison of BSGP training and inference on the `boston` dataset**
> - Using ARD kernel and ACD (with Wishart prior) kernel
>
> - Values are reported in seconds as mean ± standard deviation across three different folds.
>
> |                  | ARD                | ACD                |
> |------------------|--------------------|--------------------|
> | **Training time**  | 2082.57 ± 16.61    | 2513.56 ± 3.14    |
> | **Inference time** | 0.12 ± 0.00        | 0.13 ± 0.00       |

---

> ### Author Response · Authors · 2025-02-12
> **W3: Why infer covariate couplings simultaneously with regression? As mentioned by the authors, a standard method to work with high-dimensional data is to project it on their principle components via Principle Component Analysis. PCA extracts the whitening without costly Bayesian inference.**
>
> Many thanks for this comment, which allows us to further motivate the importance of our contribution. PCA is indeed useful to extract the principal components of the inputs and reveal some dependence/independence structure for the input covariates; however, this analysis is limited in that it does not reveal such a structure in light of how the labels are distributed conditioned on the inputs.
>
> This effect is nicely illustrated in Figure 8, where we report a synthetic experiment testing the ability of our model to recover some ground-truth couplings. We set up this experiment by generating inputs from standard normals and labels from a GP with a given $\Lambda$ matrix within a squared-exponential ACD covariance function. PCA alone here would reveal that all covariates are independent, but labels have been generated taking into account some structure in the inputs, and our approach is able to recover this. In essence, we uncover covariate couplings which are instrumental in obtaining an accurate modeling of the labels.
>
> Having said that, the way we employ PCA in this work is a working hypothesis to obtain a parsimonious parameterization. Indeed, previous works consider $\Lambda = W^{\top} W$ where $W$ is a $d \times D$ matrix to project inputs into a $d$ dimensional space. However, this requires dealing with $O(d D)$ parameters. Instead, we propose to use the parameterization $\Lambda = P_d \Lambda_d P_d^{\top}$ where $P_d$ are the $d$ principal components of the inputs, so that our parameterization requires dealing with the $d \times d$ matrix $\Lambda_d$.
>
> As a final note, it is interesting to observe that with our parameterization, even a diagonal $\Lambda_d$ would induce a full $\Lambda$, so the question is what is the advantage of using a full matrix $\Lambda_d$. We carried out one experiment to quantify the improvement offered by the added flexibility in the parameterization of $\Lambda_d$.

---

> ### Author Response · Authors · 2025-02-12
> **W4: The experimental section has no proper hypotheses. For example: "ACD with prior x outperforms ARD." or "ACD with prior x induces higher sparsity levels than ARD." with corresponding evidence/ statistical analysis. Section 5.1 reports observations in a handwavy fashion.**
>
> Thanks for the great comment. While we do not propose hypotheses along with formal hypothesis tests, we do study several hypotheses in the experiments empirically. We hypothesize that the more flexible kernel family should allow for better fits and predictive performance, which we demonstrate in section 5.1. We also hypothesise in Sec 5.2. And Sec 5.5. that we can infer covariate coupling distributions that are representative of the true underlying system. Figure 7 shows how the method can infer highly regular couplings from human motion, as expected. Finally, we present a series of ablations in sec 5.3. and 5.4. where we study the effect of sparsification and low-rank magnitudes on the resulting model. We assume that sparsification has a sweet-spot in the “moderate” regime, while low-rank approximations have a trade-off between runtime and performance.

---

> ### Author Response · Authors · 2025-02-12
> **W5: The experiments leave the question of whether sparsity is desired unanswered. The future work on statements about conditional independence of inputs seems relevant here.**
>
> This is an interesting question that can be approached from two perspectives. We assume that the true generative process of the data can be described in terms of the covariance couplings, which is the main assumption of graphical modelling literature, and has also been studied in statistics via precisions and partial correlations. In terms of model performance, we observe that the ACD itself leads to improved predictive performance, but there seems to be a tradeoff in the sparsity of the precision matrix. The horse-shoe ablation in Fig 2 shows that for different datasets different amounts of sparsity lead to best predictive performance, which is likely indicative of the true underlying data structure. We then propose three benefits to carefully tuned sparsity: it empirically improves predictive performance, it can lead to more well-specified models (in the sense of matching the underlying data structure), and it can lead to improved interpretability. As future work, we will attempt to learn the adequate level of sparsity for a given problem; this could be done by carrying out inference for the relevant sparsity-controlling (hyper)-parameters in the various priors.

---

> ### Author Response · Authors · 2025-02-12
> **[C1, C2]: Provide analytical or empirical evidence that ACD GPs are more effective in identifying couplings than classical data analysis (PCR). Carve out the differences and advantages between ACD and PCA+ARD.**
>
> Figure 8 in the appendix shows this rather well on simulated data. In this experiment, input covariates are generated from independent Gaussian distributions, so their covariance/precision does not lead to any interesting observations on covariate couplings. However, by generating labels as noisy realizations of a latent function drawn from a Gaussian process with ACD covariance, we are able to recover the corresponding couplings.
>
> In addition to Figure 8, we prepared a new figure (Figure 17) on the motion capture data where we report the inverse of the sample covariance of the covariates. We can directly compare this with the covariate couplings resulting from the GP model with the ACD covariance, observing richer insights stemming from the use of label information.
> We emphasized this point in Section 6.5 and Appendix B. See also our response to **[W3]**.

---

> ### Author Response · Authors · 2025-02-12
> **C3: Provide computation times.**
>
> We added a direct time comparison on wall clock runtime between ACD and ARD covariances for one data set (Table 5 in the appendix), and one sub-section on computational complexity (Section 4.4). See also our response to **[W2]**.

---

> ### Author Response · Authors · 2025-02-12
> **Q1: Features obtained from PCA immediately justify ARD because they are mutually orthogonal. Why should we need full-precision matrices?**
>
> Good point, which is related to the PCA-based parameterization of the ACD covariance function when (see also our response to **[W3]**). Even though the ARD parameterization after PCA yields a full $\Lambda$ matrix, the benefit of an ACD parameterization after PCA is the added flexibility, as shown in the update version of Figure 6. The added flexibility of the PCA+ACD parameterization compared to the PCA+ARD one is evident in the case where we retain most of the principal components, which leads to a nearly-diagonal precision $\Lambda$ unable to reveal any interesting couplings. We comment on these points in Sections 4.3.2 and 6.4.

---

> ### Author Response · Authors · 2025-02-12
> **Q2: Cf. Section 5.4. Are the non-diagonal couplings in lower-rank ACD matrices just projection errors incurred by cutting the rank of the PCA, thus leaving some variance unexplained?**
>
> Interesting point. In Figure 6 we show the effect of drastically reducing the number of PCA components in the ACD parameterization. The result of severely cutting information contained in the input covariance matrix, limits the possibility of discovering couplings among covariates. So our interpretation is that the unexplained variance contains useful information for revealing covariate couplings, and we should set the rank $\Lambda$ based on a tradeoff between computational and statistical efficiency. We commented on this in the revision in Section 6.4.

---

### Decision · Action_Editor_Yezz · 2025-02-05

**Recommendation:** Reject

**Comment:**

The paper discusses how relevance of coupling between covariates in GP models can be estimated using a combination of sparsity inducing priors and Bayesian inference.

All reviewers share similar concerns that are largely about the presentation of the work. While novelty as such is not an evaluation criterion, it is still important for the reader to be able to identify which elements are considered new contributions and which are from past literature, for accurate scientific communication. The paper was found lacking in this respect. It has a separate background section, but large parts of Section 3 presenting the method itself are also from previous literature and it was not always easy to identify which. Moreover, the paper was considered to be more of a tutorial on how to use these techniques, which would be appropriate for TMLR as well, but this should be better communicated for the reader. It is to some extent transparent in the writing, with sentences like "we revitalize ...", but not sufficiently so. Overall, the main purpose of the paper remains unclear, which limits the interest of the readers.

The technical content appears sound and some of the empirical findings are interesting, and there were no major comments on the technical content. The reviewers had some remarks that I believe could be addressed relatively easily, but the authors did not provide any response to the reviews. This may be partly because the review process was severely delayed in the earlier stages and I would like to apologise for the authors on behalf of TMLR. Nevertheless, the paper cannot be accepted in its current form due to the outstanding remarks.

I would encourage the authors to resubmit the work, either to TMLR or another suitable venue, after working on the main message. Try to make it easier for the readers to identify the original sources for the technical elements that are from previous literature, formulate concrete claims and recommendations for the practitioners, and choose whether the paper is a tutorial or proposes new methods. Given that there are no substantial limitations in the methodological work or the experiments, it would then be publishable.

**Audience:**

The topic itself is relevant for the audience reading TMLR, but the findings were not considered to be sufficiently interesting. The paper largely re-visits and extends some previous techniques, but lacks clear message or focus that would make it interesting for the readers, even though there are isolated details that could be of interest for some.

**Claims And Evidence:**

The authors relatively clearly tell which technical elements they want to communicate for the readers ("introduction and analysis of the model" and "alternative priors and an inference scheme"), but the paper does not make concrete claims beyond "empirical demonstration of usefulness" or provide clear hypotheses for the experiments. Consequently, it lacks accurate and convincing evidence, even though strictly speaking not violating the evaluation criterion. For example, the empirical experiments communicate the effect of the prior well, but somewhat in an anecdotal manner and the paper leaves the scientific conclusions (e.g. whether sparsity is desired) open.

**Resubmission Of Major Revision:**

The authors may consider submitting a major revision at a later time.

---

> ### Author Response · Authors · 2025-02-12
> **Response to AE**
>
> Dear Action Editor,
>
> Many thanks for handling our submission to TMLR and apologies for the delay in the response. We totally understand your position expressed in the meta-review and we hope to clarify ours.
>
> We had a revision almost ready after two reviews were submitted, but we then received a third review. In light of that, we waited to respond and prepared a revision of the paper taking into account all points raised by the third reviewer as well, including a response to her/his review.
>
> We were slightly delayed in completing the revision due to AISTATS reviewing and the ICML deadline. We have a revised version of the paper ready now (with changes highlighted in blue), which we would be happy to share (we don’t seem to be able to update the submission now). In the meantime, we respond to the reviewers below, hoping that it is possible for you to consider this revision as a resubmission of a rejected paper.
>
> We are looking forward to hearing from you soon about the next steps.
>
> Best wishes,
>
> The Authors